# Automatically Learning Feature Crossing from Model Interpretation for Tabular Data

## Abstract

Automatically feature generation is a major topic of automated machine learning. Among various feature generation approaches, feature crossing, which takes cross-product of sparse features, is a promising way to effectively capture the interactions among categorical features in tabular data. Previous works on feature crossing try to search in the set of all the possible cross feature fields. This is obviously not efficient when the size of original feature fields is large. Meanwhile, some deep learning-based methods combines deep neural networks and various interaction components. However, due to the existing of Deep Neural Networks (DNN), only a few cross features can be explicitly generated by the interaction components. Recently, piece-wise interpretation of DNN has been widely studied, and the piece-wise interpretations are usually inconsistent in different samples. Inspired by this, we give a definition of interpretation inconsistency in DNN, and propose a novel method called CrossGO, which selects useful cross features according to the interpretation inconsistency. The whole process of learning feature crossing can be done via simply training a DNN model and a logistic regression (LR) model. CrossGO can generate compact candidate set of cross feature fields, and promote the efficiency of searching. Extensive experiments have been conducted on several real-world datasets. Cross features generated by CrossGO can empower a simple LR model achieving approximate or even better performances comparing with complex DNN models.

## 1 Introduction

Recently, automated machine learning (AutoML) has been widely studied, and is proven a practical and promising machine learning technique (Yao et al., 2018). AutoML aims to provide a easy way to apply machine learning technique, and automate the procedure of machine learning partly or thoroughly. This reduces labour on data preprocessing, feature engineering, model selection, hyper-parameter tuning, model training and performance evaluation.

Among different components of machine learning, the quality of features plays an extremely important role (Liu & Motoda, 1998; Domingos, 2012). Accordingly, to improve the performances of machine learning tasks, various automatic feature generation methods have been proposed (Chapelle et al., 2015; Katz et al., 2016; Zhang et al., 2016; Qu et al., 2016; Juan et al., 2016; Blondel et al., 2016; Guo et al., 2017; Lian et al., 2018; Luo et al., 2019). Among these feature generation methods, feature crossing, which takes cross-product of sparse features, is a promising way to capture the interactions among categorical features and is widely used to enhance the performance of machine learning on tabular data in real-world applications (Chapelle et al., 2015; Cheng et al., 2016; Luo et al., 2019).

Previous works on feature crossing mostly try to search in the set of all the possible cross features (Rosales et al., 2012; Chapelle et al., 2015; Katz et al., 2016). Based on these methods, AutoCross (Luo et al., 2019) employs engineering optimization tricks, and accelerates the searching process. However, search-based methods are not efficient when the size of original feature fields is large. Meanwhile, some deep learning-based methods combine deep neural networks and various interaction components (Qu et al., 2016; Cheng et al., 2016; Wang et al., 2017; Guo et al., 2017; Lian et al., 2018). However, due to the existing of Deep Neural Networks (DNN), which can capture variety of

interactions between original features, only a few cross features can be explicitly generated by the interaction components.

As mentioned above, DNN is a powerful method for capturing various feature interactions in its hidden layers (Guo et al., 2017). This means, DNN can generate features implicitly, but can not provide interpretable cross features explicitly. Recently, the interpreration of deep models has drawn great attention in academia, and mostly focus on piece-wise interpretation, which means assigning a piece of local interpretation for each sample (Bastani et al., 2017; Alvarez-Melis & Jaakkola, 2018; Chu et al., 2018). This can be done via the gradient backpropagation from the prediction layer to the feature layer. Usually, local interpretations of a specific feature are inconsistent in different samples. And large interpretation inconsistency indicates that, the corresponding feature has interacted with other feature fields in the hidden layers of DNN. Therefore, the interpretations of DNN can help us find useful cross features.

Accordingly, in this paper, we propose a novel method called CrossGO, which can directly and automatically learn useful cross feature fields from the interpretation of DNN. (1) First, we train a common sparse DNN with the whole training set. (2) Then, for each sample in the validation set, we calculate the gradients of the output predictions with respect to the input features, i.e., local interpretations. We also calculate the global interpretation of a specific feature as the average value of all the corresponding local interpretations. For a specific feature in a specific sample, if the corresponding local interpretation is far from the corresponding global interpretation, i.e., interpretation inconsistency is large, we regard this feature interacted with others by DNN in the sample. Thus, we obtain a set of feasible features that works for feature crossing in each sample. (3) Then, we employ a greedy algorithm to generate a global candidate set containing both second-order and high-order cross feature fields. (4) With the the candidate set, we can train a simple LR model, therefore rank and select feature fields according to their contribution measured in the validation set. So far, we can obtain the final set of cross feature fields.

We have conducted experiments on several real-world datasets. The size of candidate set of cross feature fields is constrained to be $2N$, where $N$ is the number of original feature fields. To be noted, $2N$ is extremely small compared to the whole set of second-order and high-order cross feature fields, especially when $N$ is large. And a simple LR model empowered by the final set of cross feature fields achieves approximate or even better performances comparing with DNN on all datasets. This whole process of learning feature crossing can be done via simply training a DNN model and a LR model. This shows that, our proposed CrossGO method can automatically generate a compact candidate set of cross feature fields, and efficiently find enough useful cross features.

## 2 MOTIVATION

In this section, we first summarize the advantages of feature crossing and the drawbacks of conventional methods. Then we give a brief analysis about why and how we learn feature crossing from the interpretation of DNN.

### 2.1 FEATURE CROSSING

According to the definition in previous works (Cheng et al., 2016; Luo et al., 2019), we can conduct feature crossing as

$$g_{x_{f_1,i},x_{f_2,j},...,x_{f_N,m}} = x_{f_1,i} \otimes x_{f_2,j} \otimes ... \otimes x_{f_N,m}, \tag{1}$$

where $\otimes$ denotes cross-product, $g_{x_{f_1,i},x_{f_2,j},...,x_{f_N,m}}$ is the corresponding generated cross feature, and $x_{f,i}$ is a binary feature associated with field $f$, e.g., feature "gender=male" associated with field "gender". Via feature crossing, the cross feature of a male English-speaking student can be denoted as ("gender=male" and "language=English" and "job=student"). Moreover, considering feature discretization has been proven useful to improve capability of numerical features (Liu et al., 2002; Kotsiantis & Kanellopoulos, 2006; Chapelle et al., 2015), we can conduct feature discretization on numerical feature fields to generate associated binary features.

Via learning feature crossing, several advantages can be achieved: (1) new features can be generated; (2) cross features, instead of latent embeddings in deep neural networks, are highly interpretable (Luo et al., 2019); (3) simple linear machine learning methods can take use of cross features to

achieve approximate or even better performances comparing with complex nonlinear methods (Luo et al., 2019); (4) it is flexible and suits for large-scale online tasks (Cheng et al., 2016).

First attempts on feature generation mainly focus on generating second-order features (Rosales et al., 2012; Cheng et al., 2014; Chapelle et al., 2015; Juan et al., 2016; Katz et al., 2016). In (Chapelle et al., 2015), the authors try to generate and select second-order cross features according to conditional mutual information. The main problem of conditional mutual information is that, once the mutual information of an original feature is high, the generated cross features containing it will also have high conditional mutual information. ExploreKit (Katz et al., 2016) presents a framework for generating candidate features and ranking them. The proposed ranking and selecting methods are mainly based on the corresponding performances in classifiers, which is not efficient when the feature set is large. Moreover, these conventional methods fail to capture powerful high-order features.

Nowadays, deep learning-based prediction methods have shown their effectiveness. Among these methods, some try to generate and represent features implicitly or explicitly via designing various interaction components. The Wide & Deep model (Cheng et al., 2016) directly learns parameters of manually designed cross features in the wide component. The Product-based Neural Network (PNN) (Qu et al., 2016) applies inner-product or outer-product to capture second-order features. The Deep & Cross Network (DCN) (Wang et al., 2017) designs a incremental cross structure to capture second-order as well as high-order features. Factorization Machine (FM) (Rendle, 2010; 2012; Juan et al., 2016) is a suitable and success way to capture second-order feature interactions, as well as high-order feature interactions (Blondel et al., 2016). Thus, FM has been extended to deep architecture, e.g., DeepFM (Guo et al., 2017) and xDeepFM (Lian et al., 2018). As we know, these deep learning-based methods usually consists of DNN. And DNN is able to capture variety of feature interactions in its hidden layers, while not able to generate interpretable cross features. Thus, only a few cross features can be generated by the interaction components in deep learning-based methods.

Another practical method for feature generation is to employ explicit search strategies to find useful features (Fan et al., 2010; Kanter & Veeramachaneni, 2015; Katz et al., 2016; Luo et al., 2019). As we can imagine, in search-based methods, the candidate set of cross feature fields is inevitable to be extremely large, thus the searching efficiency is usually low. To solve this problem, AutoCross (Luo et al., 2019) presents a framework to search in the large candidate set more efficiently, which is a greedy and approximate alternative: (1) AutoCross iteratively searches in a set of cross feature fields, where the set is initialized as all the second-order feature fields, and the selected cross feature field is greedily used to generate new high-order cross feature fields in the next iteration; (2) AutoCross uniformly divides the dataset and validate the contribution of a cross feature field on a batch of data, which will face the problem of data lacking and random resulting when the candidate set is large. The authors apply the generated cross features, containing both second- and high-order cross features, in a LR model, and it achieves approximate or even better performances comparing with the complex DNN model in 10 real-world datasets. However, AutoCross still searches for useful cross feature fields in a large candidate set, whose size shows exponential relation with the number of the original feature fields. For example, when the size of original feature fields is 10, the number of second-order cross feature fields is 45. And when the size of original feature fields becomes 100, 200, 500 and 1000, the number of second-order cross feature fields becomes 4950, 19900, 124750 and 499500 respectively. When the candidate set is large, the searching efficiency will still be low, and data for verifying the contribution of a candidate cross feature field will be too little to produce reliable results. This strongly constrains the applicability of AutoCross in real-world applications.

## 2.2 LEARNING FROM THE INTERPRETATION OF DNN

To solve above problems of existing methods, we need to automatically generate a compact and accurate candidate set, so that the search procedure can be performed efficiently. The widely used DNN model has shown to be capable of capturing various feature interactions in its hidden layers (Guo et al., 2017). And various works has done to conduct piece-wise interpretation of DNN, which means a DNN model can be regarded as a combination of an infinite number of linear classifiers (Bastani et al., 2017; Alvarez-Melis & Jaakkola, 2018; Chu et al., 2018). Via the gradient backpropagation from the prediction layer to the feature layer, for a specific feature $x_{f,i}$ with the feature field $f$ and a specific sample $k$ in the dataset, we have

$$w_{f,i,k} = \frac{\partial \hat{y}_k}{\partial x_{f,i}}, \tag{2}$$

Table 1: The values of interpretation inconsistency of features, where $\alpha \in \{0, 1\}$ and $\beta \in \{0, 1\}$, on four toy datasets, characterizing logical operations AND, OR, XNOR and XOR respectively. Large interpretation inconsistency values give hints about cross features.

| sample | | AND | | OR | | XNOR | | XOR | |
|---|---|---|---|---|---|---|---|---|---|
| $\alpha$ | $\beta$ | $\alpha$ | $\beta$ | $\alpha$ | $\beta$ | $\alpha$ | $\beta$ | $\alpha$ | $\beta$ |
| 0 | 0 | 0.0002 | 0.0002 | 0.0001 | 0.0000 | 0.0105 | 0.0152 | 0.0137 | 0.0103 |
| 0 | 1 | 0.0006 | 0.0004 | 0.0003 | 0.0005 | 0.0000 | 0.0003 | 0.0411 | 0.0192 |
| 1 | 0 | 0.0000 | 0.0002 | 0.0002 | 0.0000 | 0.0126 | 0.0145 | 0.0189 | 0.0103 |
| 1 | 1 | 0.0000 | 0.0001 | 0.0002 | 0.0002 | 0.0308 | 0.0161 | 0.0190 | 0.0065 |

where $\hat{y}_k$ denotes the prediction made by DNN for the sample $k$, and $w_{f,i,k}$ is the local weights computed via gradient backpropagation.

**Definition 2.1** *(Local Interpretation) Given a specific feature $x_{f,i}$ and a specific sample $k$, the corresponding local interpretation is*

$$l_{f,i,k} = w_{f,i,k}\, e_{f,i}^{\top}, \tag{3}$$

*where $e_{f,i}$ denotes the corresponding feature embedding in sparse DNN.*

Usually, local interpretations of a specific feature are inconsistent in different samples in the dataset.

**Definition 2.2** *(Global Interpretation) Given a specific feature $x_{f,i}$, the corresponding global interpretation is*

$$l_{f,i} = \bar{w}_f\, e_{f,i}^{\top}, \tag{4}$$

*where $\bar{w}_f$ is the averaged local weights of features associated with the feature field $f$ in all samples, named as global weights.*

**Definition 2.3** *(Interpretation Inconsistency) Given a specific feature $x_{f,i}$ and a specific sample $k$, the corresponding interpretation inconsistency is*

$$d_{f,i,k} = \left| (w_{f,i,k} - \bar{w}_f)\, e_{f,i}^{\top} \right|. \tag{5}$$

When a specific feature has interacted with other features in the hidden layers of DNN, the corresponding local interpretations will be very inconsistent among different samples in a dataset, i.e., interpretation inconsistency is large. Thus, the interpretation inconsistency in DNN is able to lead us to generate compact and accurate candidate sets of cross feature fields. Accordingly, we can automatically learn feature crossing based on Assumption 2.1.

**Assumption 2.1** *Once the interpretation inconsistency of features in a specific feature field exceeds a threshold, the corresponding feature field works for generating cross features.*

To verify Assumption 2.1, we conduct empirical experiments on four toy datasets. The four datasets characterize four different logical operations: AND, OR, XNOR and XOR. And we have two input feature fields, where $\alpha \in \{0, 1\}$ and $\beta \in \{0, 1\}$. Thus, for each toy dataset, we have four different samples, and the output lies in $\{0, 1\}$. As we know, the logical operations AND and OR are easy and linearly separable, therefore no cross features are needed. In contrast, the logical operations XNOR and XOR are not linearly separable, therefore second-order cross feature field consisting of $\alpha$ and $\beta$ should be generated for promoting linear classifiers. We train a two-layer DNN model on the four toy datasets until convergence, where the Area Under Curve (AUC) evaluation becomes 1.0. On all datasets, the gradients from the prediction layer to the feature layer is computed, and the interpretation inconsistency is obtained, as shown in Table 1. It is clear that, interpretation inconsistency values on AND and OR are extremely small, while those on XNOR and XOR are relatively large. This observation clearly shows that, $\alpha$ and $\beta$ should be crossed on XNOR and XOR, while should not on AND and OR. This is clearly consistent with common sense, and strongly proves Assumption 2.1. According to the values in Table 1, 0.01 might be a proper threshold for filtering feature fields to construct compact and accurate candidate sets of cross feature fields. So far, we can conclude that, it is proper to learn feature crossing from the interpretation inconsistency computed in DNN.

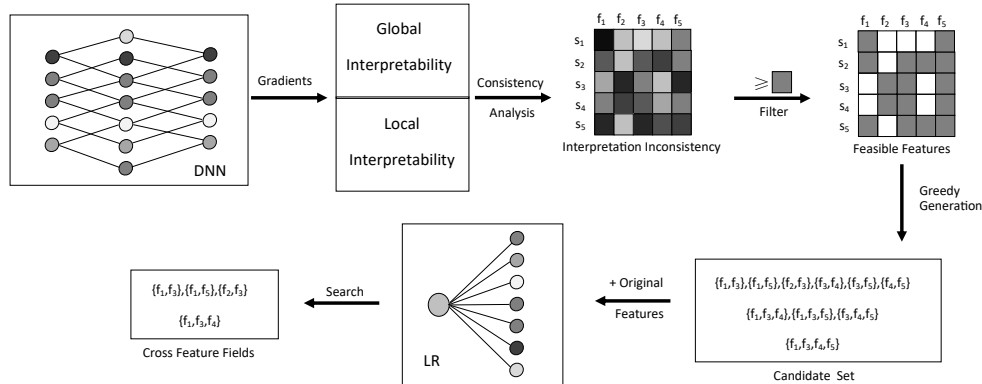

Figure 1: Overview of our proposed CrossGO approach. $s_i$ and $f_j$ indicate the i-th sample and the j-th original feature field respectively.

## 3 CROSSGO

In this section, we formally propose the CrossGO model. In general, CrossGO consists of two steps: (1) generating a compact and accurate candidate set of cross feature fields; (2) searching in the candidate set for the final cross feature fields. We use $\{f_1, f_2, ..., f_n\}$ to denote the cross feature field generated by crossing $f_1, f_2, ..., f_n$. Figure 1 provides an overview of the proposed CrossGO approach, which will be described in detail below.

### 3.1 CANDIDATE SET GENERATION

Firstly, as we rely on the piece-wise interpretation of DNN to generate the compact and accurate candidate set of cross features, we need to train a DNN model. The input is the original categorical features and categorical features are field-wise high dimensional vectors. Thus, we use embedding layer (Zhang et al., 2016) to transform the input features into low dimensional dense representations. Then the dense representations are passed through some linear transformation and nonlinear activation to obtain the predictions of samples, where we use relu as the activation for hidden layers, and sigmoid as the activation for the output layer to support binary classification tasks.

Based on the trained DNN model, according to Assumption 2.1, in every validation sample $k$, we first compute the interpretation inconsistency $d_{f,i,k}$ of each feature $x_{f,i}$, where $d_{f,i,k}$ is defined in Definition 2.3. Thus, we obtain a matrix $D$, where $D_{kf}$ is the interpretation inconsistency of f-th feature in the k-th sample. Then we conduct element-wise filter on matrix $D$ with a threshold $\eta$ which is the threshold in Assumption 2.1. And $\eta$ is 0.01 for default. The filter operation can be formulated as

$$D'_{kf} = \begin{cases} 1, & D_{kf} \geq \eta \\ 0, & otherwise \end{cases}, \tag{6}$$

where $D'$ is a binary feasible feature matrix. According to Assumption 2.1, for each feasible feature $D'_{kf} = 1$, the corresponding feature field $f$ can be used to generate candidate cross feature fields.

Lastly, we perform greedy generation to obtain the candidate set of cross features fields. Pseudocode of greedy generation is given in Algorithm 3.1. Here, we introduce three new hyper-parameters: $\delta$, $\varepsilon$ and $\gamma$. $\delta$ is the maximum order of cross features, and $\delta$ is 4 for default, where higher-oder cross features are rarely useful. $\varepsilon$ denotes the threshold of the occurrence ratio of cross feature fields on validation set, and $\varepsilon$ is 0.01 for default. $\gamma$ is the number of candidate cross feature fields returned by greedy generation, and $\gamma$ is $2N$ for default. To be noted, in general, $2N$ is extremely small comparing with the size of the whole set of 2nd-order, 3th-order and 4th-order cross feature fields. For example, $N$ is 100, the size of the whole set is $4087875$ ($4950 + 161700 + 3921225$).

---

**Algorithm 1** Greedy Candidate Set Generation

---

**Require:** feasible feature matrix $D'$, set of original feature fields $F = \{f_1, f_2, ..., f_N\}$, maximum order $\delta$ of cross features, threshold $\varepsilon$ of the occurrence ratio of cross feature fields on validation set, number $\gamma$ of candidate cross feature fields, the size of validation set is $M_{valid}$.

**Ensure:** compact and accurate candidate set $S$.

1: $S=\{\}$, $S'=\{\}$, set of first-order features $S_1 = \{\{f_1\}, \{f_2\}, ..., \{f_N\}\}$, current order $o = 2$, count of occurrences $count_c = 0$ for every cross feature field $c$;
2: **for** $o \leq \delta$ **do**
3:    $S_o = \{\}$, $P = \{\}$;
4:    **for** sample $k$ in validation set **do**
5:        set of feasible feature fields $Q_k = \{f| D'_{kf} = 1\}$;
6:        **for** $f$ in $F$, $c$ in $S_{o-1}$ **do**
7:            generate cross feature field $p = \{f\} \cup c$;
8:            **if** $p \subset Q_k$ **then**
9:                $count_p = count_p + 1$;
10:               $p \rightarrow P$;
11:           **end if**
12:       **end for**
13:   **end for**
14:   **for** $p$ in $P$ **do**
15:       **if** $count_p/M_{valid} \geq \varepsilon$ **then**
16:           $p \rightarrow S_o$;
17:       **end if**
18:   **end for**
19:   $S' = S' \cup S_o$;
20:   $o = o + 1$;
21: **end for**
22: select top $\gamma$ cross feature fields with largest counts from $S'$, and assign them to $S$;
23: **return** $S$.

---

## 3.2 SEARCHING FOR CROSS FEATURE FIELDS

After obtaining the candidate set of cross feature fields $S = \{c_1, c_2, ..., c_{2N}\}$, where $c_i$ is a specific cross feature field, we can search for the final useful cross feature fields according to their contribution measured in the validation set.

We first need to train a LR model. The input of LR model consists of both original features and candidate cross features. We use $S$ as the schema to process training data to generate training cross features, which are denoted as $X^{cross}_{train}$. The shape of $X^{cross}_{train}$ is $[M_{train}, 2N]$. We also denote the original features of training set as $X^{original}_{train}$, and the shape of it is $[M_{train}, N]$. Thus, the number of model's input feature fields is $3N$. Therefore, the time cost of training this LR model is in the same order of magnitude with directly training a LR model with only original features.

Based on the trained LR model with model weights $W$, we conduct our searching process. Here, we use $X^{original}_{valid}$ and $X^{cross}_{valid}$ to denote the original features and the cross features in the validation set respectively. Pseudocode of searching process is given in Algorithm 3.2. It is important to note that the step 4-10 of Algorithm 3.2 can be conveniently paralleled via multi-threading or multi-process implementation. Moreover, the inference process is much faster than the training process, and therefore the time cost of searching can be neglected.

This searching process is fast and effective. The reason why we can use such a easy searching strategy is, CrossGO can generate a compact and accurate candidate set of cross feature fields according to the interpretation inconsistency in DNN.

---

**Algorithm 2** Searching for Cross Feature Fields on Validation Set.

---

**Require:** candidate set $S = \{c_1, c_2, ..., c_{2N}\}$, original features $X_{valid}^{original}$ with $N$ feature fields
   and $M_{valid}$ samples, candidate cross features $X_{valid}^{cross}$ with $2N$ cross feature fields and $M_{valid}$
   samples, labels $Y_{valid}$ with $M_{valid}$ samples, model weights $W$ of LR trained with both orig-
   inal and cross features on training set, sigmoid function $\sigma()$ and AUC computing function
   $compute\_auc()$.
**Ensure:** final set of cross feature fields $S^*$.
 1: $S^* = \{\}$;
 2: $b(-1) = X_{valid}^{original} W^\top[0 : N - 1]$;
 3: $AUC(-1) = compute\_auc(Y_{valid}, \sigma(b(-1)))$;
 4: **for** $i$ in $[0, 2N - 1]$ **do**
 5:    **for** $j$ in $[0, 2N - 1]$ **do**
 6:       **if** $c_j$ not in $S^*$ **then**
 7:          $b(j) = b(-1) + X_{valid}^{cross}[:, j] W[N + j]$;
 8:          $AUC(j) = compute\_auc(Y_{valid}, \sigma(b(j)))$;
 9:       **end if**
10:    **end for**
11:    $k = \underset{j}{\operatorname{argmax}} AUC(j)$;
12:    **if** $k \geq 0$ **then**
13:       $c_k \rightarrow S^*$;
14:       $b(-1) = b(k)$;
15:       $AUC(-1) = AUC(k)$;
16:    **else**
17:       break;
18:    **end if**
19: **end for**
20: **return** $S^*$.

---

Table 2: Statistics of the benchmark datasets.

| Datasets | #Samples | | #Feature Fields | | Domain |
|----------|----------|---------|---------|---------|--------|
|  | Training | Testing | #Num. | #Cate. | |
| Employee | 29,494 | 3,277 | 0 | 9 | Human Resource |
| Adult | 32,562 | 16,282 | 6 | 8 | Social |
| Allstate | 131,823 | 56,497 | 15 | 115 | Insurance |
| Prudential | 41,567 | 17,816 | 22 | 104 | Insurance |
| Movielens | 588,799 | 150,215 | 109 | 20 | Entertainment |
| Criteo | 41,256K | 4,584K | 13 | 26 | Advertising |
| Anon1 | 233,123 | 58,282 | 178 | 15 | Securities |
| Anon2 | 99,137 | 42,487 | 459 | 26 | Banking |

## 4 EXPERIMENTS

In this section, we empirically test our proposed crossGO model on 8 datasets for thorough com-
parisons. We first describe the datasets and settings of the experiments, then report and analyze the
experimental results.

### 4.1 DATASETS

We evaluate the proposed crossGO model on six benchmark and two anonymous datasets. The two
anonymous datasets are provided with sanitization and named as Anon1 and Anon2. The statistic of
these datasets are shown in Table 2. All the tasks are for binary classification. Employee, Adult and
Criteo are the same datasets as in (Luo et al., 2019). For other datasets except for the former three
and Movielens, we use the first 70% data as the training data, while the rest 30% as testing. And we
cut out the last 20% of the training data as validation set.

Table 3: Summary of results in terms of AUC (in percent) on **narrow datasets**.

|          | LR    | CMI   | FM    | AutoCross | DNN   | CrossGO   |
|----------|-------|-------|-------|-----------|-------|-----------|
| Employee | 86.75 | 89.01 | 86.89 | 89.42     | 87.85 | **89.59** |
| Adult    | 92.10 | 91.53 | 91.71 | 92.80     | 92.68 | **92.85** |
| Criteo   | 78.55 | 78.52 | 79.27 | 80.34     | 79.85 | **80.41** |

Table 4: Summary of results in terms of AUC (in percent) on **wide datasets**.

|            | LR    | CMI   | FM    | AutoCross | DNN   | CrossGO   |
|------------|-------|-------|-------|-----------|-------|-----------|
| Movielens  | 81.49 | 82.61 | 85.73 | 85.12     | 86.51 | **86.77** |
| Prudential | 84.56 | 84.78 | 84.88 | 84.82     | 84.95 | **84.97** |
| Allstate   | 86.10 | 86.35 | 86.41 | 86.41     | 86.63 | **86.73** |
| Anon1      | 72.36 | 73.11 | 72.54 | 72.67     | 74.60 | **76.23** |
| Anon1      | 89.19 | 89.91 | 90.02 | 89.33     | 90.89 | **91.03** |

For Movielens, we regard reviews as behaviors, and sort the reviews from one user by time. For user $u$, we split $u$'s behaviors into three parts: the first 50% is for training, following 20% is for validation, while the rest is for testing. Meanwhile, we transform Movielens into a binary classification data. Specifically, original user rating of the movies is continuous value ranging from 0 to 5. We label the samples with rating of 4 and 5 to be positive and the samples with rating of 1 and 2 to be negative. We extract a series of statistical features based on each user's historical behavior. Features include user_age, user_gender, user_occupation, is_cate, recent_k_cate_mean_score and recent_k_cate_cnt, where k $\in \{10, 30, 50\}$ and cate $\in \{$action, adventure, animation, children's, comedy, crime, documentary, drama, fantasy, film-noir, horror, musical, mystery, romance, sci-fi, thriller, war, western$\}$. The recent_k_cate_cnt is the count of each category in movies that each user has reviewed in recent $k$ times. Accordingly, the recent_k_cate_mean_score is the average score of each category in movies that each user has reviewed in recent $k$ times.

In summary, in terms of whether or not the number of original feature fields is bigger than 100, we test our approach on two types of datasets:

- **Narrow Datasets**: Employee, Adult and Criteo;
- **Wide Datasets**: Movielens, Allstate, Prudential, Anon1 and Anon2.

## 4.2 EXPERIMENTAL SET-UP

As our goal is to make a simple LR model empowered by generated cross features to achieve approximate or even better performances comparing with DNN, we compare against the strong baselines (LR, CMI+LR and DNN) and the state-of-the-art feature crossing approach (AutoCross+LR) as in (Luo et al., 2019). We simplify AutoCross+LR and CMI+LR to AutoCross and CMI. Accordingly, our approach is named as CrossGO.

We use the same discretization method as in (Luo et al., 2019) for preprocessing data. For the DNN model, which we rely on to compute the interpretation inconsistency of features: learning rate is 0.001, the dimensionality of embedding vector is 16, network structure is 64-32. For the LR model, which we rely on to search for cross features: learning rate$\in [0.0001, 0.001]$. For all models, we use Adam with weight decay as zero as optimizer, the Area-Under-Curve (AUC) as our experimental metric and the same early-stopping strategy. Specifically, if the validation AUC dose not increase in three epochs, the training process will be stopped.

## 4.3 PERFORMANCE ANALYSIS

Results of our experiments are summarized in Table 3, Table 4, Table 6 and Figure 2. According to Table 3 and Table 4, all models except CMI significantly beat LR model on all datasets. CMI is inferior to LR on some datasets, which shows that adding new cross features does not guarantee the improvement of performance. The set generated by CMI may contain many redundant and over-fitting cross features, which leads to poor performance on some datasets.

Table 5: Comparison of cross feature generation time (in seconds) on each dataset.

| | Employee | Adult | Criteo | Movielens | Prudential | Allstate | Anon1 | Anon2 |
|---|---|---|---|---|---|---|---|---|
| AutoCross | 182 | 128 | 11094 | 2829 | 452 | 1595 | 6136 | 8144 |
| CrossGO | 59 | 62 | 5922 | 367 | 86 | 196 | 161 | 288 |

Table 6: The number of second/high-order cross feature fields generated on each dataset.

| | | Employee | Adult | Criteo | Movielens | Prudential | Allstate | Anon1 | Anon2 |
|---|---|---|---|---|---|---|---|---|---|
| AutoCross | 2nd-order | 7 | 6 | 8 | 52 | 75 | 62 | 68 | 120 |
| | high-order | 3 | 1 | 5 | 2 | 0 | 3 | 6 | 14 |
| CrossGO | 2nd-order | 12 | 3 | 8 | 150 | 156 | 133 | 117 | 237 |
| | high-order | 3 | 7 | 7 | 9 | 0 | 21 | 39 | 121 |

Obviously, our proposed CrossGO model performs better than DNN on all datasets. That is to say, a simple LR model empowered by our generated cross features always achieves better performance comparing with DNN. Moreover, Figure 2 demonstrates that we only need a small portion of our generated cross feature fields to make LR model achieving approximate DNN performance. And as continuing adding our generated cross feature fields, the performances of LR with CrossGO are increasing. Thus, user can decide the number of generated cross feature fields to add according to their expectation of testing performance. To be noted that as mentioned in section 3, the number of generated cross feature fields has little impact on the time cost of CrossGO. In summary, these results strongly verifies the effectiveness, efficiency and flexibility of our approach.

Meanwhile, our proposed approach outperforms AutoCross on all datasets. On the narrow datasets, the results in Table 3 demonstrate that AutoCross is inferior to our proposed CrossGO model. On the wide datasets, the candidate set of cross features is inevitable to be extremely large, and samples for evaluating each candidate cross feature field are too little. Thus, AutoCross performs poorly on wide datasets, and can not achieve similar perfromances as DNN. Our porposed CrossGO model can significantly outperform AutoCross on wide datasets. Moreover, according to Table 5, CrossGO is more efficient than AutoCross, especially on wide datasets. And wider the dataset, larger our advantage.

## 4.4 INTERPRETABILITY OF CROSSGO

As CrossGO can explicitly generate cross features, CrossGO has good interpretability. In order to make the interpretability of CrossGO comprehensible, we choose a representative case from dataset MovieLens in the process of testing. The five most important cross feature fields added to the final set are {is_horror, recent_50_action_cnt}, {is_action, recent_50_crime_mean_rate}, {is_horror, recent_50_thriller_mean_rate}, {is_children's, recent_30_fantasy_cnt} and {is_action, recent_30_crime_mean_rate}. Obviously, these five cross feature fields are complete comprehensible to human. And from a logical point of view, they are highly likely to make data more linear-separable. For example, {is_horror, recent_50_action_cnt} implies that for predicting current rating, it is a good choice to consider both whether current film is horror and whether current user often watches action films. And {is_action, recent_50_crime_mean_rate} implies that it is also a good choice to consider both whether current film is action and whether current user likes crime films. In practice, adding these cross features significantly improves the performance of LR model. This demonstrates that our proposed CrossGO model is interpretable and effective.

## 5 CONCLUSION

In this paper, we define the interpretation inconsistency in DNN for the first time. According to the interpretation inconsistency, we propose a novel CrossGO method. CrossGO can generate a compact candidate set of cross feature fields, with extremely small amount compared to the whole set of second-order and high-order cross feature fields in a dataset. Based on the corresponding performances in the validation set, useful cross feature fields can be directly ranked and selected from our compact candidate set. The whole process of learning feature crossing can be done via simply training a DNN model and a LR model. Extensive experiments have been conducted on several real-world datasets. Cross features generated by CrossGO can empower a simple LR model

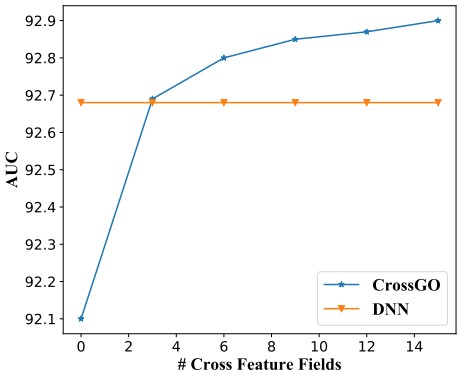 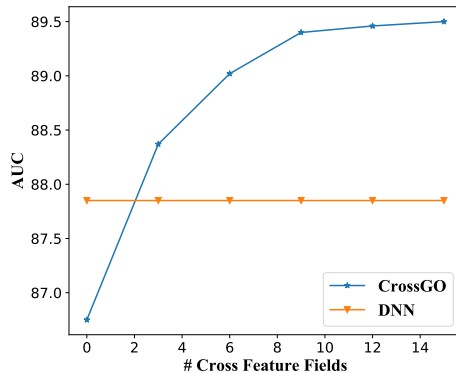

(a) Testing performances on Adult w.r.t. the number of our generated cross features.

(b) Testing performances on Employee w.r.t. the number of our generated cross features.

Figure 2: Testing performances on Adult and Employee w.r.t the number of our generated cross features.

achieving approximate or even better performances comparing with complex DNN models, as well as the state-of-the-art feature crossing method, i.e., AutoCross, which greedily explores the whole set of cross feature fields. Moreover, cross features generated by CrossGO have great interpretability.

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
