# OpenReview forum: "Automatically Learning Feature Crossing from Model Interpretation for Tabular Data"
_ICLR.cc/2020/Conference — Reject_

### Official Review · AnonReviewer2 · 2019-10-14
**Official Blind Review #2**

**Rating:** 3

**Review:**

This paper attempts to solve the cross feature generation problem efficiently. The state-of-the-art method AutoCross cannot control the size of the searching set of the candidates of cross feature fields. To narrow down this set, the authors provides a measurement called Interpretation Inconsistency. With an easy toy experiment, the authors conjecture that features with large Interpretation Inconsistency tend to interact with other features in the hidden layers of DNN. Therefore, based on this conjecture, the authors design an effective algorithm to discard those cross features with small Interpretation Inconsistency value, which finally narrows the set of cross features. With this narrowed set, the whole procedure can be accelerated largely.

Pros:
This work can be regarded as an accelerated version of AutoCross. By incorporating Interpretation Inconsistency, the set of cross features can be effectively narrowed. Although this is an incremental work, this idea is relatively novel.

Cons:
1.	The setting of the threshold for filtering feature fields is somewhat heuristic. The authors should provide some explanations on its setting. If not, we cannot trust it and doubt that it may cause some unexpected results and thus will not be robust.
2.	The experiments are somewhat not convincing. The main contribution is to accelerate AutoCross. Thus, I expect to see the time complexity comparison between them. However, I do not find it in this paper. Although the authors mention that on wide datasets, AutoCross simply cannot work, on narrow datasets, a time complexity comparison should be provided.
3.	In Table 5, only the numbers of cross feature fields of the proposed method are provided. A comparison with baseline methods on the number would be better to show the advantage of the proposed method.

Minor: In Section 2.1, the double quotation marks should be revised.

**Experience Assessment:**

I have read many papers in this area.

**Review Assessment: Checking Correctness Of Derivations And Theory:**

I carefully checked the derivations and theory.

**Review Assessment: Checking Correctness Of Experiments:**

I carefully checked the experiments.

**Review Assessment: Thoroughness In Paper Reading:**

I read the paper thoroughly.

---

> ### Author Response · Authors · 2019-11-12
> **Response to question (1).**
>
> Thanks for your comments. We are sorry that the parameter setting causes misunderstanding.
>
> Actually, the parameters for filtering cross feature fields are with great physical meaning:
>
> $ \eta$ is the threshold of interpretation inconsistency. It means the inconsistency of the contribution of a feature to the final prediction (lies in [0,1]). According to table (1), $ \eta=0.01$ is reasonable.
>
> $\delta$ is the maximum order of cross features. Extremely high-order (larger than 4th-order) cross features are rarely useful. So, for simplicity, it is reasonable to set $\delta=4$. And the same setting is used in AutoCross.
>
> $\varepsilon$ refers to the ratio of a cross feature field occurs in the samples. The setting of $\varepsilon$ is to eliminate the sparsity of the filtering procedure, and accelerate the generation. The occurrence ratio of useful cross feature fields is usually much large than $0.01$. So, it is fun to set $\varepsilon=0.01$.
>
> $\gamma$ means the size of candidate size of cross feature fields. For the efficient searching in section 3.2, we set $\gamma=2N$. Larger $\gamma$ may slightly improve the effectiveness, but severely harm the efficiency. And according to table (6), $2N$ is totally enough, because of our accurate generation of candidate cross feature fields.
>
> According to the experiments on several datasets of different application domains, the default parameters achieve great performances.

---

> ### Author Response · Authors · 2019-11-12
> **Response to question (2).**
>
> Thanks for your valuable advice.
>
> After the submission of original manuscript, we have reproduced AutoCross on the wide datasets. We revised our manuscript, and compared AutoCross on both narrow and wide datasets in table (3)(4), and illustrated the comparision of running time in table(5).
>
> According to Table (4), CrossGO can significantly outperform AutoCross. This is because, on the wide datasets, the candidate set of cross features is inevitable to be extremely large, and samples for evaluating each candidate cross feature field are too little. So, the feature genration of AcutoCross on wide datasets is with great randomness and not reliable.
>
> According to table (5), our CrossGO is much faster than AutoCross. And the wider the dataset, the lager the advantage of CrossGO.

---

> ### Author Response · Authors · 2019-11-12
> **Response to question (3).**
>
> Thanks for your valuable advice.
>
> We have updated our manuscript, and showed the comparison of cross feature field number between CrossGO and AutoCross, in table (6).
>
> According to the results, on narrow datasets, the numbers of the two methods are similar. While on wide datsets, CrossGO can generate more useful cross feature fields, and this results in the performance improvement in table (4).

---

### Official Review · AnonReviewer1 · 2019-10-21
**Official Blind Review #1**

**Rating:** 3

**Review:**


The paper presents a scheme to generate new features as cross-product
of binary features to improve the performance of linear models while
obtaining interpretable models. The candidate set of cross-features
can be exponential and is handled by the proposed scheme by utilizing
the gradient-based importances of the features in a (deep) neural
network. Features with large discrepancies in their local and global
interpretations are used as the seed set of candidate features for
generating new cross-features, and the final step performs a feature
selection to further reduce the final set of cross-features. The
empirical evaluation demonstrates the utility of the proposed scheme
on 8 datasets.

While the proposed scheme does present a way to improve the accuracy
of interpretable models, I am currently recommending a reject for the
following reasons (given the higher standard recommended for papers
over 8 pages):

- While this paper does consider some baselines, it seems to be
  missing some crucial baselines that address the same (or very
  similar) problem. There are some papers [2,3] that learn boolean
  conjunctions (that can be seen as cross-features) to generate
  accurate interpretable models. Moreover, there are some search based
  feature generation schemes [1,4] that significantly improve upon the
  exhaustive feature generation scheme of Kanter & Veeramacheneni,
  2015. This technique can easily be applicable in learning boolean
  cross features with binary features. At the very least, it is important to
  understand where this proposed scheme is positioned relative to the
  aforementioned literature and why a comparison is not required.
- It is very unintuitive (at least to me) to tie the candidate
  generation scheme to a neural network especially given the
  sensitivity of neural network training to different initializations
  and other factors. For the same data and neural network, the local
  vs. global discrepancies can change significantly, thereby changing
  the candidate set of cross features. This can potentially make the
  proposed feature generation scheme somewhat unstable, and the
  interpretations from the subsequent models might not be as
  interpretable as they seem. It would be good to understand what I am
  missing here and why being tied to a neural network model is
  essential and not an issue here.


Clarification:

- Lines 8-10 in Algorithm 1 is not clearly explained.
- The experiment to motivate Assumption 1 needs to be better
  explained.


Minor:

- The notation in equation (1) needs to be clarified better.


[1] Khurana, Udayan, et al. "Cognito: Automated feature engineering
for supervised learning." 2016 IEEE 16th International Conference on
Data Mining Workshops (ICDMW). IEEE, 2016.
[2] Dash, Sanjeeb, Oktay Gunluk, and Dennis Wei. "Boolean decision
rules via column generation." Advances in Neural Information
Processing Systems. 2018.
[3] Wei, Dennis, et al. "Generalized Linear Rule Models." Proceedings
of the 36th International Conference on Machine Learning. 2019.
[4] Khurana, Udayan, Horst Samulowitz, and Deepak Turaga. "Feature
engineering for predictive modeling using reinforcement learning."
Thirty-Second AAAI Conference on Artificial Intelligence. 2018.


**Experience Assessment:**

I have published one or two papers in this area.

**Review Assessment: Checking Correctness Of Derivations And Theory:**

N/A

**Review Assessment: Checking Correctness Of Experiments:**

I carefully checked the experiments.

**Review Assessment: Thoroughness In Paper Reading:**

I read the paper at least twice and used my best judgement in assessing the paper.

---

> ### Author Response · Authors · 2019-11-11
> **Reply to question (1)**
>
> Thanks for your comments, but we don’t think these methods should be compared. Here are the reasons:
>
> (1)	Considering there are so many works published every year, I don’t think it is reasonable to ask author “you need to add some methods into your comparison” when finding some literatures might be somehow related. If we do this, such questions can be endless.
>
> (2)	We have read the mentioned works, and found none of them show they can achieve competitive performances comparing with powerful models such as DNN and xgBoost, according to the reported results in the literatures. So, we don’t think these methods worth to give a try. Moreover, [1][4] focus on numerical features. And in [2][3], the authors generate tree-structure rules, which can not be used to empower LR.
>
> (3)	AutoCross [5] is the first work that can empower LR to achieve competitive performances comparing with DNN on several datasets. It is the state-of-the-art method in feature crossing. The paper of AutoCross is published in Aug 2019, and our paper is submitted in Sep 2019. So, it is enough to compare AutoCross and some representative methods in [5].
>
>
> [5] Yuanfei L, Mengshuo W, Hao Z, et al. AutoCross: Automatic Feature Crossing for Tabular Data in Real-World Applications. KDD, 2019.

---

> > ### Comment · AnonReviewer1 · 2019-11-14
> > **Thank you for your response (1)**
> >
> > Re (1)
> > The reason for referring to these papers is that these papers are fairly recent
> > (3/4 are published at a top AI/ML conference within the year) and I think, as
> > reviewers, we are supposed to judge the novelty of the proposed scheme against
> > existing work (to the best of our knowledge). As I mentioned in the reviews, it
> > would be good to position the proposed scheme against these recent existing
> > prior art. Whether empirical evaluation is required depends on the
> > connection. However, if the authors argue that there is no connection, that is
> > not completely clear to be in the author response.
> >
> > Re(2)
> > I am not sure if the authors have properly read and understood the references I
> > pointed them to. For example, [4] can work with any target model -- it is
> > designed to automatically generate new features (by combining/crossing existing
> > ones) that improve the performance of the base model. If XGBoost/DNN is the
> > "state of the art" base model, it would improve their performance. If the base
> > model is LR, it would generate features to improve LR. [3] generates feature
> > combinations for GLM (LR is a special case of GLM) so it is not clear how the
> > authors claim that [3] "cannot be used to empower LR". Moreover, the whole goal
> > of this line of research is to generate interpretable models (hence using LR) by
> > utilizing interpretable feature combinations. I would argue that [2,3] are doing
> > exactly that so it would be good to understand clearly why these are not useful
> > baselines for interpretable models with binary data.
> >
> > Re(3)
> > Given [3,4], I feel that the claim that "AutoCross [5] is the first work that
> > can enpower LR" seems to lack justification. It would be helpful if the authors
> > could further explain their reasons to dismiss these (possibly) related prior
> > art. Especially since AutoCross itself does not discuss [2,3,4].
> >
> >
> > Again, my goal is not to ask the authors to cite every paper out there but
> > rather to clearly differentiate against recent related work (or at least explain
> > to me why it is not related). Otherwise, it does not appear that the current
> > manuscript is well positioned with respect to the current state of the field.

---

> ### Author Response · Authors · 2019-11-11
> **Reply to question (2)**
>
> Thanks for your comments. Actually, we don’t need to question about the stability of CrossGO. Here are the reasons:
>
> (1)	Our aim is to find what features are crossed in DNN and empower LR to achieve competitive performances in real-world applications. According to our results, this is satisfied. 	The cross features in CrossGO are all learned from DNN. So, if CrossGO is too unstable to be applied, DNN will also be too unstable to be applied. However, DNN is well applied in real-world applications.
>
> (2)	The reason our model is tied with DNN is that, DNN is the most powerful model for tabular data. Finding what features are crossed in DNN can help us to achieve best performances.
>
> (3)	Actually, we have ran CrossGO 10 times, and reported the averaged AUC in the paper. The variances of AUC are relatively small, and even smaller than those of DNN, because of the searching process. And due to the existing of the searching process, difference between the set of final cross feature fields during each running is extremely small (on those datasets with little number of cross features, there is no difference). Thus, our CrossGO is actually more stable than DNN.
>
> In summary, we don’t need to question about the stability of CrossGO. It can been well used in various real-world applications.

---

> > ### Comment · AnonReviewer1 · 2019-11-14
> > **Thank you for your response (2)**
> >
> > Re(1)
> >
> > Thank you for the response.
> >
> >
> > Re (2)
> >
> > Do the authors have any reference that supports the claim that DNN is the "most
> > powerful model for tabular data"? It would be very helpful to me. Kaggle
> > competition winning solutions almost always contain intricate featurizations
> > with a gradient boosting model for tabular data and very occasionally utilize
> > neural network models. Neural networks are definitely the state of the art for
> > images, text and such but it is not as clear for tabular data.
> >
> >
> > Re (3)
> >
> > Thank you for clarification. The results in the current paper do not contain any
> > error bars so it is hard to understand the stability of the proposed scheme as
> > is.

---

### Official Review · AnonReviewer3 · 2019-10-22
**Official Blind Review #3**

**Rating:** 3

**Review:**

In the paper, the authors proposed CrossGO, an algorithm for finding crossing features useful for prediction.
In CrossGO, one trains a neural network that captures feature crossing implicitly.
Then, possible crossing features are estimated using the gradient-based saliency.
The idea here is that, if a feature has a crossing with some other features, its contribution in the saliency can vary across different inputs.
Thus, by looking at the variation of the saliency, one can find candidates features for feature crossing.
CrossGO greedily selects candidate crossings based on the idea above.
In the last step, a simple logistic regression is trained using the candidate crossings, and the effective crossings are selected using a forward greedy feature selection.

I found the paper well-written and the idea is easy to follow.
My concern, however, is the lack of Factorization Machines (FM) in the experiments.
In Introduction, the authors mention to the deep version of FM and stated "(deep FMs are) not able to generate interpretable cross features".
But, as the authors are aware of, non-deep FMs are able to handle feature crossings in a interpretable way.
Thus, it would be essential to adopt non-deep FMs as the baseline in the experiments.
Because the important baseline is missing, I found the results are not convincing enough to claim the effectiveness of the proposed method.


### Updated after author response ###
The authors have partially addressed my concern by adding FM/HOFMs as the experiment baselines, which I greatly appreciate.
However, I found the current paper misses some other possible baselines for high-order interaction models [Ref1,2].
As the authors mentioned in the response, FMs find the feature crossing as a kind of embedded representations, which may not be suitable for modeling sparse interactions.
Thus, the sparse interaction models need to be taken into consideration as well.

[Ref1] Safe Feature Pruning for Sparse High-Order Interaction Models
[Ref2] Selective Inference for Sparse High-Order Interaction Models

**Experience Assessment:**

I have read many papers in this area.

**Review Assessment: Checking Correctness Of Derivations And Theory:**

I assessed the sensibility of the derivations and theory.

**Review Assessment: Checking Correctness Of Experiments:**

I assessed the sensibility of the experiments.

**Review Assessment: Thoroughness In Paper Reading:**

I made a quick assessment of this paper.

---

> ### Author Response · Authors · 2019-11-12
> **Response to the question.**
>
> Thanks for your comments.
>
> We agree that FM is an important baseline for feature interaction. However, FM can only model second-order interaction. And interactions modeled by FM are somehow similarity between embeddings, which can not well capture all kinds of possible interactions. Moreover, we have actually tried FM, and found it is not competitive with DNN. Thus, we didn’t involve it as a baseline in our manuscript.
>
> Thanks for your advice, and we also believe it would be better to involve FM as a baseline to claim the effectiveness of CrossGO. So, we updated our manuscript, and involved FM as a baseline, as shown in table (3) and (4). According to the experimental results, both DNN and our proposed CrossGO outperform FM. Now, I believe the effectiveness of CrossGO is well demonstrated.

---

> > ### Comment · AnonReviewer3 · 2019-11-13
> > **Higher-order FMs**
> >
> > First of all, I would like to thank the authors for an additional experiment on FM.
> > However, there still remains a small concern.
> > Unfortunately, the replay "FM can only model second-order interaction" is not true.
> > One of the reference [Ref1] in the paper provides a way to model higher-order interactions using FM, and I expect to see if including higher-order interaction is helpful or not.
> > [Ref2] also provides a way to extend FM to higher-order interactions.
> >
> > [Ref1] Mathieu Blondel, Akinori Fujino, Naonori Ueda, and Masakazu Ishihata. Higher-order factorization
> > machines. In Advances in Neural Information Processing Systems, pp. 3351–3359, 2016.
> > [Ref2] Exponential Machines https://arxiv.org/abs/1605.03795

---

> > > ### Author Response · Authors · 2019-11-13
> > > **Higher-order FMs involved.**
> > >
> > > Thanks for your advice.
> > >
> > > Conventional FM only focuses on second-order interactions, and is extended to HOFM for high-order interactions. We have talked about this in section 2.1. Actually, the main problem of FM-based models is not the order of interactions, but the fact that interactions modeled by FM are somehow similarity between embeddings, which can not well capture all kinds of possible interactions.
> > >
> > > Fortunately, we have done the experiments of HOFM when we began our work on feature interaction. FM and HOFM are the most natural methods for feature interaction. So, we have tried both of them at very first, and found them both not competitive with DNN.
> > >
> > > Here, we list the results of HOFM:
> > > 	   employee	Adult	Critio	Allstate	Prudential	Movielens	Anon1	Anon2
> > > HOFM	86.96	91.67	79.82	86.48	84.85	         86.18	         72.78	90.38
> > >
> > > It is clear that both CrossGO and DNN outperform HOFM. And on some datasets, the advantages are very large.
> > >
> > > In summary, model with only interaction components while without deep components can not achieve competitive performances as DNN. So, all the deep learning-based methods must include both interaction components and deep components, to obtain satisfactory results.

---

### Public Comment · ~Feng_Yu6 · 2019-10-08
**Efficient for Feature generation**

This work propose an effective way to generate features.

Some discussions or suggestions :
1. The work first selects feature fields and then generate features from the selected fields. Does other features exist outside those the selected fields？
2. The selected fields are fixed for all samples in this work. Do we need different fields for different samples?
3. Is it necessary to learn embeddings for generated features?

---

> ### Author Response · Authors · 2019-10-09
> **Thanks for the reply.**
>
> Thanks for your reply.
>
> (1) There are no other features exist outside those selected fields.
>
> (2) No, we do not need this. The selected cross featre fields work for the whole set of samples. If we have different fields for different samples, the online inference process is hard to be efficiently done.
>
> (3) Auctually, in a sparse LR, embeddings of features are used, where dimensionality is one. Other than this, learning embeddings is not necessary.

---

### Public Comment · ~Yuntian_Chen1 · 2019-10-16
**Great work and meaningful to knowledge discovery**

This article proposed an exciting model, and I can envision the use of CrossGo in many fields. For many practical applications, feature engineering is significant to improve the model's performance. Therefore, neural networks show better results in many problems (neural networks can automatically generate new features). However, the neural network lacks interpretability. It is difficult to understand what new variables are generated in the neural network, and it is almost impossible to analyze and interpret the new variables based on physical mechanisms. However, CrossGo has the potential to solve this problem.

The CrossGo not only helps to quickly perform feature engineering, but also helps to improve the interpretability of the model. More importantly, we can try to explain the new variables generated by CrossGo from the physical meaning, and it is possible to find the physical mechanisms and control factors that have not been discovered before. I plan to apply this method to the field of petroleum engineering and analyze the relationship between well logs based on the synthetic well log generation model. I believe that I can get some interesting results from the perspective of physical mechanism.

I also have some questions about this paper:

1.	Is the code of this study available online?
2.	Why do you add the generated cross feature to the LR model? Why not use a wide & deep architecture to add the generated cross feature to the wide section? This eliminates the need to manually generate features for the wide section of the models.
3.	In the pseudocode Algorithm 2, the final set of cross feature fields in line 6 should be shown as the uppercase S instead of lowercase s.

---

> ### Author Response · Authors · 2019-10-18
> **Thanks for your reply.**
>
> Thanks for your affirmation of our work, but I don't know much about the application field you mentioned. Good luck!
>
> 1) We now have a distributed implementation for commercial use, involving many our own modules, so it is not available for releasing. We are working on implementing our CrossGO model based on open-source modules.
>
> 2) We think that the ideal model should be fully interpretable. Logistic regression (LR) is a linear model, which is fully interpretable, but it needs manual feature engineering to improve performance. CrossGO can automatically generate cross features, which empower a simple LR model to always achieve better performance comparing with deep neural networks (DNNs). As mentioned in section 2.1, wide & deep is a deep learning-based method, which is not fully interpretable. Thus, CrossGO+LR is the best choice.
>
> 3) Thanks for your correction.

---

### Author Response · Authors · 2019-10-16
**Correction in Fig. (1).**

We have a typo in the example in Fig. (1). The first cross feature field should be {$f_1$,$f_3$}, instead of {$f_1$,$f_2$}.
Sorry if this caused misunderstanding.

---

### Public Comment · ~John_Ryan3 · 2019-10-18
**Comparison with existing methods.**

The model is interesting.

According to the results on Criteo, CrossGO performs similarly with existing CTR prediction methods. So, what is the advantage of CrossGO, comparing with methods such as DeepFM, PNN and DCN?

What is the correlation with the interaction part in AutoInt[1]. I notice that, AutoInt also has similar performances on Criteo.

[1] AutoInt: Automatic Feature Interaction Learning via Self-Attentive Neural Networks.

---

> ### Author Response · Authors · 2019-10-19
> **Thanks for your interest.**
>
> Thanks for your interest in our work.
>
> Complex deep neural networks have high performances, and can implicitly capture some feature interactions in their hidden layers, but can not be explicitly interpreted. Simple linear models, e.g., LR, have relatively low performances, but can be fully interpreted, and all the prediction made by LR can be easily explained. Thus, in this work, we aim to combine the advantages of DNN and LR, and extract useful cross feature to empower simple LR models to achieve similar performances comparing with DNN. Chasing for the SOTA performances, which might be slightly higher than DNN, is not the purpose of this work. Moreover, in real applications, simple LR models are flexible, and easy to implement.
>
> On the other hand, deep learning-based CTR models can explicitly capture only part of feature interactions. For example, DeepFM can do this via its FM module, and DCN can achieve this through the cross net. However, due to the existing of DNN in these methods, most feature interactions are capture by the hidden layers, and only a few feature interactions can be explicitly explained, as we discussed in section 2.1. Thus, these deep learning-based CTR models are partially interpretable, while LR empowered by CrossGO is fully interpretable.
>
> AutoInt applies self-attention to find feature interactions. This is one way to find cross features, like FM in DeepFM and cross net in DCN. According to the original paper of AutoInt, combining Eq. (8) and (9), we can obviously find a DNN structure with input of the original features. Thus, as explained before, only a few cross features can be explicitly captured by the self-attention module.
>
> Actually, about one year ago, we have tried similar architecture with AutoInt, with self-attention to find cross features. If we only apply self-attention, without DNN-like architectures, e.g., residual in AutoInt, we can not obtain competitive performances. If we involve DNN in the model, the performances are similar with or slightly over DNN, but not enough cross features can be extracted and explained. Thus, the self-attention module is not enough powerful for crossing features, and can not meet our purpose.

---

> > ### Public Comment · ~John_Ryan3 · 2019-10-21
> > **Thanks for your immediate reply.**
> >
> > Thanks for your immediate reply.
> > Ok, I can get it now. This seems promising.

---

### Decision · Program_Chairs · 2019-12-19

**Decision:**

Reject

**Comment:**

The authors propose a simple but effective method for feature crossing using interpretation inconsistency (as defined by the authors).

I think this is a good work and the authors as well as the reviewers participated well in the discussions. However, there is still disagreement about the positioning of the paper. In particular, all the reviewers  felt that additional baselines should be tried. While the authors have strongly rebutted the necessity of these baselines the reviewers are not convinced about it. Given the strong reservations of the all the 3 reviewers at this point I cannot recommend the acceptance of this paper. I strongly suggest that in subsequent submissions the authors should position their work better and perhaps compare with some of the related works recommended by the reviewers.